# Laminar microcircuitry of visual cortex producing attention-associated electric fields

**Jacob A Westerberg[1]\*, Michelle S Schall[1], Alexander Maier[1], Geoffrey F Woodman[1], Jeffrey D Schall[2]**

[1]Department of Psychology, Center for Integrative and Cognitive Neuroscience, Vanderbilt Vision Research Center, Vanderbilt Brain Institute, Vanderbilt University, Nashville, United States; [2]Centre for Vision Research, Vision: Science to Applications Program, Departments of Biology and Psychology, York University, Toronto, Canada

**Abstract** Cognitive operations are widely studied by measuring electric fields through EEG and ECoG. However, despite their widespread use, the neural circuitry giving rise to these signals remains unknown because the functional architecture of cortical columns producing attention-associated electric fields has not been explored. Here, we detail the laminar cortical circuitry underlying an attention-associated electric field measured over posterior regions of the brain in humans and monkeys. First, we identified visual cortical area V4 as one plausible contributor to this attention-associated electric field through inverse modeling of cranial EEG in macaque monkeys performing a visual attention task. Next, we performed laminar neurophysiological recordings on the prelunate gyrus and identified the electric-field-producing dipoles as synaptic activity in distinct cortical layers of area V4. Specifically, activation in the extragranular layers of cortex resulted in the generation of the attention-associated dipole. Feature selectivity of a given cortical column determined the overall contribution to this electric field. Columns selective for the attended feature contributed more to the electric field than columns selective for a different feature. Last, the laminar profile of synaptic activity generated by V4 was sufficient to produce an attention-associated signal measurable outside of the column. These findings suggest that the top-down recipient cortical layers produce an attention-associated electric field that can be measured extracortically with the relative contribution of each column depending upon the underlying functional architecture.

## Editor's evaluation

By combining rare EEG and laminar recordings in monkeys, Westerberg and colleagues studied the neural correlates of the well-known attention-related N2pc signal and found that it is due to the activation of extra-granular layers of cortex. Further, this effect was stronger for columns that were more feature selective. These findings are extremely important and a unique contribution to the literature on the neurobiology of attention.

\*For correspondence:
jacob.a.westerberg@vanderbilt.edu

**Competing interest:** The authors declare that no competing interests exist.

## Introduction

Research into extracortical electric fields provides fundamental insights into the mechanisms of human perception, cognition, and intention. For instance, event-related potential (ERP) components like the N2pc (*Eimer, 1996*; *Luck and Hillyard, 1994*; *Woodman and Luck, 1999*) and Pd (*Hickey et al., 2009*) reliably index selective attention in humans and monkeys, alike. However, the interpretation

of these extracortical measures of attention is severely limited by uncertainty about the exact neural processes that generate these signals (*Nunez and Srinivasan, 2006*). Understanding what brain processes an electric field indicates requires knowing how it is generated (e.g., *Cohen, 2017*).

One avenue to localize neural generators of electric fields is through inverse source localization (*Michel et al., 2004*; *Grech et al., 2008*). However, the results are indefinite and cannot offer conclusive answers. Moreover, these methods do not allow for the probing of the underlying neural circuitry. For example, most EEG signals are hypothesized to be generated by interlaminar interactions in cortical columns (*Nunez and Srinivasan, 2006*). Columnar microcircuits are ubiquitous across the brain (*Douglas et al., 1989*; *Douglas and Martin, 1991*), having a well-defined anatomical structure (*Mountcastle, 1997*; *Kaas, 2012*) and consistent physiological activation pattern (*Bastos et al., 2012* but see *Godlove et al., 2014*). The canonical cortical microcircuit offers a framework in which to interpret columnar dynamics in sensory or cognitive tasks, yet the relationship between this functional architecture and electric fields related to cognition commonly measured in humans is unexplored.

Electric fields measured at the surface of the brain (ECoG) and scalp (EEG) are theorized to be generated by dipoles in cortex. However, measuring current dipoles requires sampling electrical potentials across all the layers of the cerebral cortex. Such laminar neurophysiological measurements are rare and unsystematic in humans. Work in rodents has uncovered intriguing insights into cortical laminar microcircuits underlying evoked EEG signals, but all of these were limited to sensory responses (*Jellema et al., 2004*; *Bruyns-Haylett et al., 2017*; *Nass et al., 2021*). Fortunately, macaque monkeys produce homologues of the attention-associated EEG signals (N2pc: *Woodman et al., 2007*; *Cohen et al., 2009*; *Purcell et al., 2013*; Pd: *Cosman et al., 2018*). Laminar neurophysiological measurements (*Schroeder et al., 1998*; *Maier et al., 2010*; *Buffalo et al., 2011*; *Hansen et al., 2011*; *Self et al., 2013*; *Godlove et al., 2014*; *Engel et al., 2016*; *Klein et al., 2016*; *Hembrook-Short et al., 2017*; *Nandy et al., 2017*; *Trautmann et al., 2019*; *Westerberg et al., 2019*; *Tovar et al., 2020*; *Ferro et al., 2021*) and EEG (*Schmid et al., 2006*; *Woodman et al., 2007*; *Sandhaeger et al., 2019*) are well established in macaques. However, despite many studies linking intra- and extracortical signals (*Schroeder et al., 1992*; *Whittingstall and Logothetis, 2009*; *Musall et al., 2014*; *Snyder and Smith, 2015*), to date, little is known about the laminar origins of ERPs in primates.

Here, we show that visual cortex generates dipoles through layer-specific transsynaptic currents that give rise to electric fields that track the deployment of selective attention. These dipoles were generated by the extragranular compartments of cortex, indicating these cognitive operations likely arise from top-down interactions. Moreover, functional architecture – in the form of feature columns – was associated with the relative contribution of individual, local cortical columns to the global electric field. These results are the first to our knowledge to describe laminar specificity in synaptic activations contributing to the generation of electric fields associated with cognitive processing.

## Results
### Attention task
To investigate extracortical manifestations of attention-associated electric fields, we trained macaque monkeys to perform a visual search task (*Figure 1A*). Three macaque monkeys (designated Ca, He, and Z) performed visual search for an oddball color target (red or green), presented within an array of five or seven uniform distractors (green or red) (N sessions for each monkey: Ca 21, He 9, Z 18). A fourth monkey (P) performed visual search for an oddball shape (T or L) presented within an array of up to seven uniform distractors (L or T) (N sessions: monkey P, 22). Each animal performed well above chance (chance level for monkeys Ca, He: 16.6%; P, Z: 12.5%) (behavioral accuracy in color search: Ca 88%, He 81%, Z 85%, shape search monkey P 66%). We sampled cortical neural signals during the color pop-out search to be certain of which item received the benefit of attention in the array. We used the more difficult search data to determine the generality of our findings. Two different recording types were used, necessitating four monkeys total, as described below and summarized in *Supplementary file 1*.

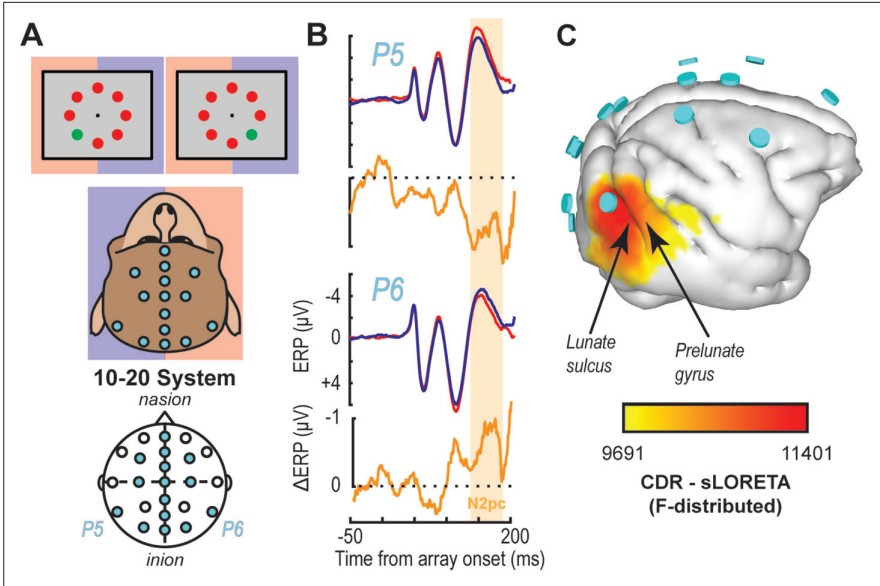

**Figure 1.** EEG traces and inverse source localization for the N2pc index of attention in monkeys. (**A**) EEG was recorded from electrodes arranged according to the 10–20 system in monkeys performing visual search by shifting gaze to a colored oddball stimulus (monitor diagrams show two example arrays). Blue and red shading highlights mapping between visual hemifield and cerebral hemisphere. (**B**) Trial-averaged P5 and P6 EEG traces from monkey Z following presentation of search arrays with the target in either the right (blue) or left (red) visual hemifield as well as the difference (orange). The voltage difference between the target in the left versus right hemifields reveals the N2pc ~150 ms after array presentation. The N2pc was significant (dependent samples t test between polarizations averaged between 125 and 250 ms after array presentation (t(35) = 2.42, p = 0.02)). (**C**) Inverse solution of current distribution consistent with difference in voltage distribution during the N2pc (113–182 ms) when the target was in the left hemifield versus right hemifield using sLORETA. Current density is displayed over the three-dimensional (3D) boundary element model derived from a magnetic resonance scan of monkey Z. Data was clipped below the 85% maximum value for display purposes. Cyan disks indicate EEG electrode positions. Current density is concentrated beneath electrode P6 caudal to the lunate sulcus and in area V4 on the prelunate gyrus. Results are reproduced for a second monkey in *Figure 1—figure supplement 1*.

The online version of this article includes the following figure supplement(s) for figure 1:

**Figure supplement 1.** N2pc distribution of monkey P (10–20 EEG recordings).

## Inverse modeling of attention-associated extracortical electric fields points to visual cortex

Once animals could perform visual search, we implanted an array of electrodes approximating the human 10–20 system in monkeys P and Z (*Figure 1A*). Using these electrodes, we observed extracortical electric dynamics in both monkeys. An index of attention known as the N2pc manifests during visual search. The N2pc electric field indexes attention allocation in this task. The magnitude of the N2pc was largest over occipital sites (*Figure 1B*, *Figure 1—figure supplement 1*), consistent with previous reports in humans and macaques (*Luck and Hillyard, 1994*; *Eimer, 1996*; *Woodman and Luck, 1999*; *Hopf et al., 2000*; *Woodman et al., 2007*; *Cohen et al., 2009*; *Purcell et al., 2013*). We used sLORETA inverse modeling for source localization. Previous source estimates for the N2pc identified the human homologue of V4 (*Luck and Hillyard, 1990*; *Luck and Hillyard, 1994*; *Hopf et al., 2000*). These findings are consistent with numerous reports that areas in mid-level visual cortex in monkeys produce robust attention signals (*Moran and Desimone, 1985*; *Luck et al., 1997a*; *McAdams and Maunsell, 1999*; *Reynolds et al., 1999*; *Fries et al., 2001*; see *Roe et al., 2012* for review) across cortical layers (*Engel et al., 2016*; *Nandy et al., 2017*). Consistent with these earlier studies, the inverse model showed that current sources include V4 on the prelunate gyrus (*Figure 1C*, *Figure 1—figure supplement 1*). However, the modeled current sources also included other cortical regions, as is common for inverse solutions. Notably, the inverse solution identifies V1 to be about as strong as V4 in contributing to the N2pc, which is unlikely given current knowledge on attentional

modulation for each area (*Motter, 1993*; *Luck et al., 1997a*; *Kastner et al., 1999*; *Buffalo et al., 2011*). Given the primary feature used in the search task was color, we investigated the laminar profile of attention-associated electric field generation in V4 where color is better represented (*Roe et al., 2012*).

## V4's laminar microcircuit produces dipoles that predict the attention-associated electric field

Guided by magnetic resonance imaging (MRI), linear multielectrode arrays (LMAs) were inserted into area V4 of monkeys Ca and He. LMAs were placed perpendicular to the cortical surface, spanning supragranular (L2/3), granular (L4), and infragranular (L5/6) cortical layers (*Figure 2—figure supplement 1*). We confirmed that attentional modulation of spiking activity could be observed during pop-out search performance consistent with previous reports (*Westerberg et al., 2020a*). Moreover, the laminar profile of attentional modulation matched that of attentional modulation in a different task with spiking activity in the middle layers being the most highly enhanced with attention (*Figure 2*; *Nandy et al., 2017*). Critically, while attentional modulation is present in the laminar data prior to the emergence of extracortical attention-associated fields such as the N2pc, that cross-laminar modulation persists through this interval.

Simultaneous with LMA recording, an extra-cortical electric signal was recorded immediately above V4 – critically the recording took place outside of the cortical column itself. Current source density (CSD) was derived from the local field potentials (LFPs) sampled across V4 layers. To relate the extracortical signal (*Figure 3A*) to synaptic currents estimated as CSD (*Figure 3B–D*), we employed information theory to capture multivariate factors and nonlinearities between signals (*Shannon, 1948*; *Cover and Thomas, 2006*). Importantly, information theory analyses are model independent (*Timme and Lapish, 2018*). Information theory is thus superior to standard linear models since these models cannot capture all potential relationships between signals. The relationship between the extracortical signal and CSD was assessed in four distinct steps, as illustrated by a representative session (*Figure 3E–F*, *Figure 3—figure supplement 1*). We use the interval of the N2pc to determine whether laminar circuitry in V4 can contribute to the attention-associated electric fields. This interval occurred before the median

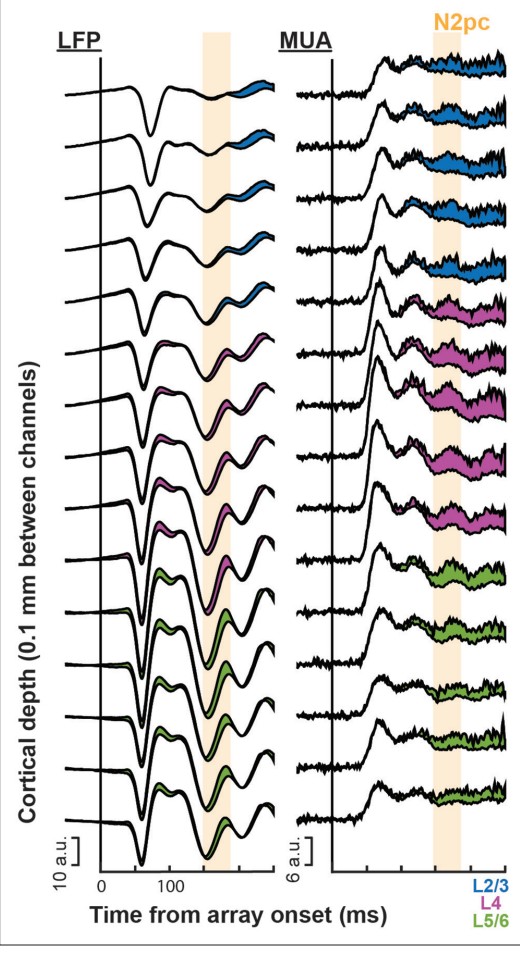

**Figure 2.** Laminar profile of local field potential (LFP) and multiunit (MUA) attentional target selection during visual search task performance across monkeys Ca and He (n = 2). Responses were averaged across sessions (n = 30) at each of the depths (n = 15) relative to the L4/5 boundary (magenta to green). Difference between target (attended) and distractor (unattended) responses represented by the fill color corresponding to the recording channels' laminar compartment. Top line of each trace combination is the attended condition, bottom trace is the unattended condition. Significant differences in magnitude of attention effect, averaged 150–190 ms after search array onset, across laminar compartment were detected through an ANOVA for both LFP (F(2, 442) = 22.43, p = 5.2e$^{-10}$) and MUA (F(2, 442) = 3.87, p = 0.022). Note the effect of attention in the MUA was largest in the middle layers (M$_{L2/3}$ = 2.68, M$_{L4}$ = 3.50, M$_{L5/6}$ = 2.38), consistent with previous reports (*Nandy et al., 2017*). Time of the N2pc as measured throughout the main text (150–190 ms following array onset) indicated with orange.

The online version of this article includes the following figure supplement(s) for figure 2:

**Figure supplement 1.** Laminar alignment and receptive field mapping.

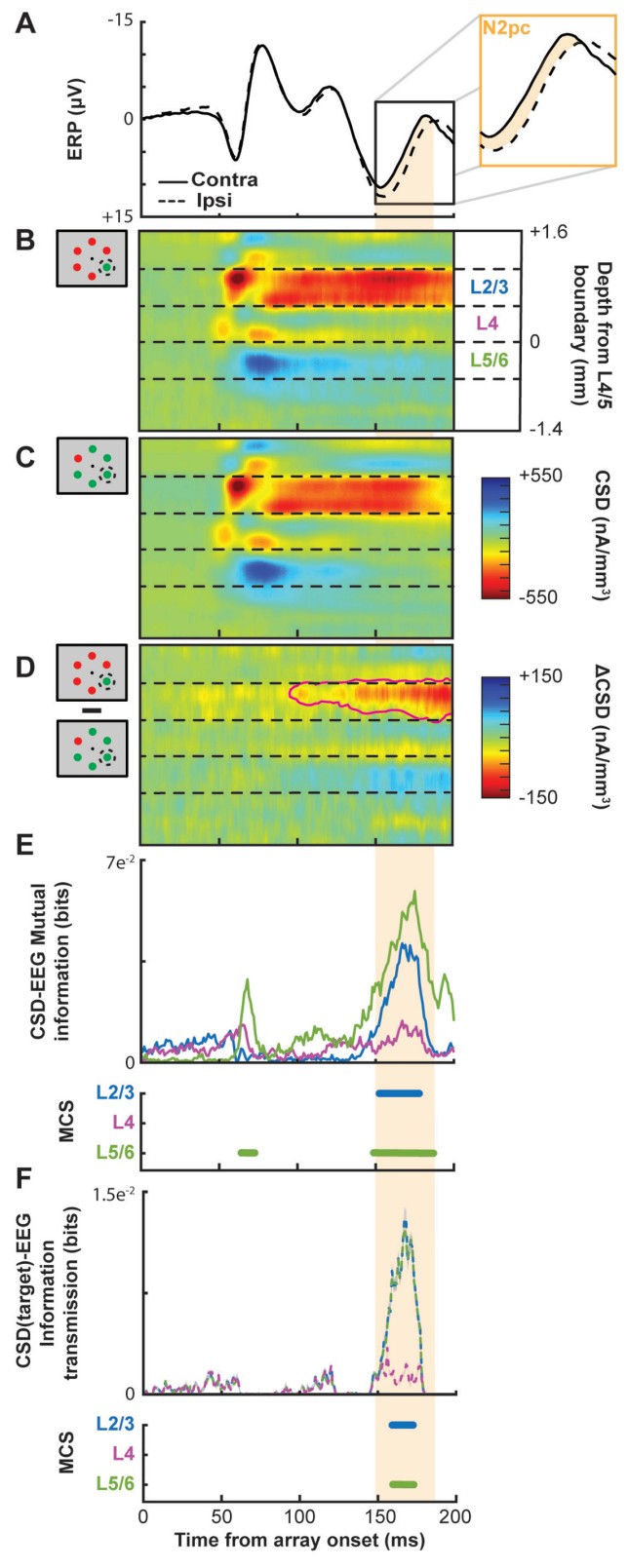

**Figure 3.** Extracortical attention-associated signal and simultaneously recorded V4 synaptic currents during representative session. (**A**) Extracortical event-related potential (ERP) voltages after search array presentation, averaged over all trials when the target was presented contra- (solid) or ipsilateral (dashed) to the electrode. Inset magnifies the N2pc interval defined as the difference in potentials 150–190 ms after the array appeared

*Figure 3 continued*

(orange highlight). (**B**) Simultaneous current source density (CSD) when the target appeared in the population receptive field of the column. Dashed lines indicate boundaries between supragranular (L2/3), granular (**L4**), and infragranular (L5/6) layers. CSD values were interpolated and smoothed along depth for display only. Current sinks have hotter hues, and current sources, cooler. The earliest sink arises in putative L4, likely from rapid feedforward transmission, followed by intense, prolonged sinks in L2/3 accompanied by weaker source in L5/6. (**C**) CSD evoked by distractor in the receptive field has similar pattern. (**D**) Subtraction of CSD responses to target versus distractor in receptive field. The only statistically significant differences (determined through a t test across time-depth with p < 0.05, outlined by magenta line) were due to a current sink in L2/3 that arose gradually ~100 ms after array presentation. This relative sink was associated with a weak relative source in L5/6. (**E**) Simultaneous mutual information between CSD and the extracortical signal for L2/3 (blue), L4 (purple), and L5/6 (green). Times with significant mutual information were computed through Monte Carlo shuffle simulations (MCS). N2pc interval is highlighted. Intervals with significant mutual information persisting for at least 10 ms are indicated by horizontal bars. No mutual information with EEG was observed in L4. (**F**) Information transmission about target position from V4 CSD to the extracortical signal. Conventions as in E.

The online version of this article includes the following figure supplement(s) for figure 3:

**Figure supplement 1.** Mutual information measures for the extracortical signal, V4 current source density (CSD), and target position.

response times for each monkey contributing laminar V4 data ([median± standard deviation]: monkey Ca 227 ± 49 ms, He 225 ± 44 ms).

First, we employed Monte Carlo simulations of the mutual information analysis to verify that the extracortical signal exhibited significantly enhanced information about target position during the time window of the N2pc. Second, we measured target information across the layers of V4 during the N2pc interval. This analysis revealed enhanced information in L2/3 and L5/6 but not in L4. Third, we computed the mutual information between the extracortical signal and CSD during the N2pc window, irrespective of target position. This analysis showed a significant relationship between the extracortical signal and the CSD in L2/3 and L5/6 but not in L4. Fourth, we measured the transmitted information about target location from CSD to extracortical signal during the N2pc interval (*Timme and Lapish, 2018*). This analysis demonstrated significant information transmission to the extracortical signal from L2/3 and L5/6, but not L4.

Averaged across sessions, we observed that the electric field during the N2pc interval (*Figure 4A*) was associated with a consistent CSD pattern (*Figure 4B*). This relationship was observed in each monkey (*Figure 4—figure supplement 1*). Presentation of the search array in any configuration elicited an early current sink in L4, followed by a prolonged sink in L2/3 that was associated with a briefer source in L5/6.

We next computed information transmission about target location from the CSD to the extracortical signal for each session (*Figure 4C*). All cortical layers provided significant information transmission in >75% of sessions during the N2pc window (150–190 ms following array onset). However, the magnitude of transmitted target information was significantly greater in L2/3 and L5/6 relative to L4 (L2/3-L4: t(29) = 2.15, p = 0.040; L5/6-L4: t(29) = 2.20, p = 0.036). The magnitude of information transmission was not significantly different between L2/3 and L5/6 (t(29) = 0.21, p = 0.84). Across sessions, the three other information theoretic analyses were consistent with the example session (*Figure 3—figure supplement 1*). Moreover, significant information transmission during the N2pc was observed in each monkey (*Figure 4—figure supplement 1*).

To verify the results, we applied the information theoretic analysis over a longer interval (*Figure 4—figure supplement 2*). Importantly, we found no signal differences or significant information transmission in the 100 ms pre-array baseline period as expected with baseline correction. We also evaluated the interval 200–250 ms following array presentation and found a polarization reversal in the extracortical signal likely corresponding to the Pd (*Cosman et al., 2018*). We observe persistent current differences in the extragranular CSD during this interval sufficient to contribute to the extracortical signal. However, we observed no statistically significant information theoretic relationship between the CSD and extracortical signal during this interval. The absence of a relationship could indicate no actual association or be a consequence of the reduced trial count due to the clipping of signals at saccade initiation. This uncertainty prevents further consideration of this interval in these data.

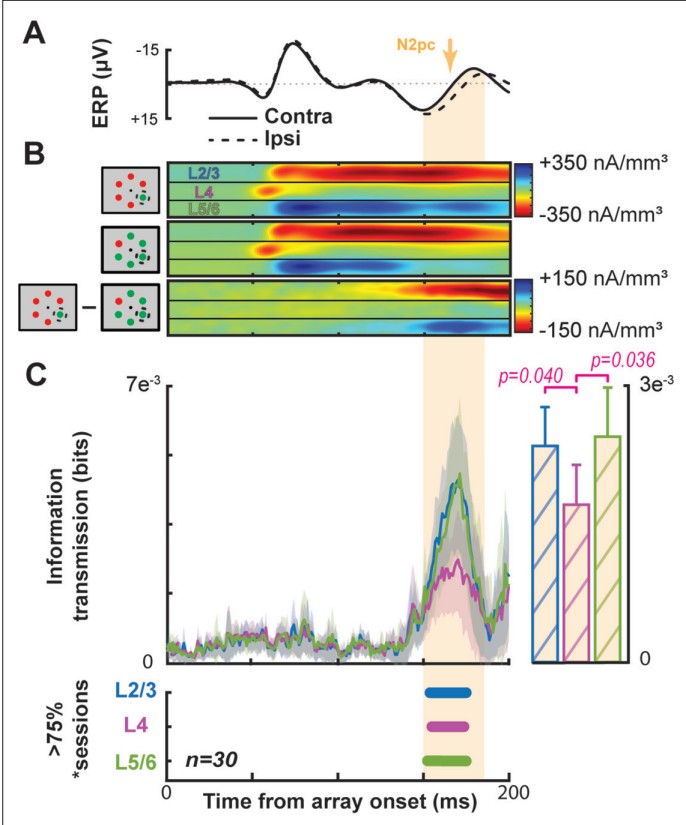

**Figure 4.** Grand average demonstrating the link between V4 current source density (CSD) and the extracortical attention-associated electric field. Conventions as in *Figure 3*. (**A**) Average event-related potential (ERP) across all sessions and animals with the target contra- (solid) or ipsilateral (dashed). The N2pc interval is indicated by orange shading. (**B**) Average V4 CSD with the target in (top) or out of the receptive field (RF) (center) with the difference between the two at the bottom. (**C**) Grand average information transmission about target position from V4 layers to the extracortical signal as a function of time (left). Average +2 SEM of information transmission during the N2pc window (right). Panel below shows that information transmission from L2/3 and in L5/6 was significantly greater than that from L4 (t test p < 0.05). Timepoints with significant information transmission were assessed through Monte Carlo simulations during >75% of sessions. Intervals with significance persisting for at least 10 ms are indicated by horizontal bars, color coded for each laminar compartment (bottom).

The online version of this article includes the following figure supplement(s) for figure 4:

**Figure supplement 1.** Individual monkey physiology and information transmission.

**Figure supplement 2.** Grand average results with expanded interval.

**Figure supplement 3.** Information theoretic relationship between V4 current source density (CSD) and extracortical signal persists when accounting for stimulus identity.

**Figure supplement 4.** Microsaccades do not explain information theoretic relationship between V4 current source density (CSD) and extracortical signal during the N2pc.

---

Last, we performed two additional analyses to determine whether the observed information theoretic relationship is confounded by spurious factors. First, we measured the contribution of V4 neuron selectivity for stimulus color. We computed information transmission separately for trials with a red stimulus and with a green stimulus in the receptive field (RF). In the population average of the two calculations for each session, we observed significant information transmission during the N2pc (*Figure 4—figure supplement 3*). Hence, the relationship between V4 activity and the EEG does not depend on color specificity. Second, we measured the contribution of microsaccades, which have been linked to attentional modulation in V4 (*Lowet et al., 2018*). We computed information transmission separately for trials without microsaccades (*Figure 4—figure supplement 4*). In the population average of the two calculations for each session, we observed

significant information transmission during the N2pc. Hence, microsaccade production was not responsible for the observed information theoretic associations between signals. The outcomes of these control analyses engender more confidence that the current dipole in V4 generated by the L2/3 CSD sink and the L5/6 CSD source contributes to the N2pc measured in the extracortical electric field.

## Columnar feature selectivity influences contribution to N2pc

Given the columnar organization of color tuning of V4 neurons (*Figure 5A*; *Roe et al., 2012*; *Zeki, 1973*; *Zeki, 1980*; *Tootell et al., 2004*; *Conway and Tsao, 2009*; *Kotake et al., 2009*), we investigated the association between the N2pc and the CSD in columns with different color preferences. To quantify color selectivity through depth, we computed the response ratio between red and green stimuli (*Figure 5B*). Responses were measured as power in the gamma range (30–150 Hz) because this signal reflects local circuit interactions (*Ray and Maunsell, 2011*) and feature selectivity in visual cortex (*Berens et al., 2008*) and is more reliably measured than spiking activity across all LMA contacts. This analysis collapses across differences in color tuning across layers, so although the interlaminar specificity of gamma activity is not fully understood, recent work indicates that laminar gamma power can reliably reflect feature selectivity in a spatially specific fashion (*Westerberg et al., 2021b*).

To identify columns with significant selectivity for either red or green, we performed Wilcoxon signed rank tests between the distribution of ratios in each column against bootstrapped null distributions. Each bootstrapped null distribution contained 15 randomly selected ratios from the full dataset (450 experimental values) from which 1000 distributions were generated. The bootstrapped distributions represent the range of possible values observed across V4, but do not capture any difference in the homogeneity of feature selectivity within a column.

We found that more than half of V4 columns show selectivity for red or green stimuli (monkey Ca 12/21 [57.1%], He 5/9 [55.6%]). We computed the information transmission of target position for each color tuned column separately for trials when the preferred or the non-preferred color was in the column's population RF. Across sessions with different target and distractor colors, we observed no difference in the amplitude of the extracortical signal during the N2pc (paired sample $t(16) = 0.40$, $p = 0.69$) (*Figure 5C*) nor the laminar CSD (L2/3: $t(16) = -0.85$, $p = 0.41$; L4: $t(16) = 0.75$, $p = 0.46$; L5/6: $t(16) = 0.36$, $p = 0.72$) (*Figure 5D*). However, information transmission during the N2pc was greater when a preferred rather than a non-preferred color was in the RF (*Figure 5G*). This difference was significant in L2/3 and L5/6 but not in L4 ($t$ test across time with at least 10 ms having $p < 0.05$) and is evident in single sessions (*Figure 5—figure supplement 1*).

We investigated whether the magnitude of information transmission varied with degree of color preference. In session-wise correlations of the difference in information transmission between preferred and non-preferred colors at the time of peak information transmission (160–180 ms) as a function of columnar color selectivity index (CCSI), we found a significant relationship for L2/3 (Spearman's $R = 0.50$, $p = 0.005$) and L5/6 ($R = 0.51$, $p = 0.004$) but not L4 (*Figure 5H*).

We also tested whether feature selective columns, on average, transmitted more information than their non-feature-selective counterparts. We found that feature selective columns, in all laminar compartments, transmitted significantly more information (*Figure 5I*) (two-sample $t$ test: L2/3, $p = 0.044$; L4, $p = 0.023$; L5/6, $p = 0.009$). As such, we wanted to determine if this was due to a lack of attentional modulation in the non-selective columns. This was not the case, we observed that non-selective columns were modulated with attention. Attentional modulation was observed in both the CSD in L2/3 and L5/6 (one-sample $t$ test: L2/3: $t(64) = -6.01$, $p = 9.8e^{-8}$; L4: $t(64) = -0.18$, $p = 0.86$; L5/6: $t(64) = 5.24$, $p = 1.9e^{-6}$) as well as across all layers in the population spiking activity (one-sample $t$ test: L2/3: $t(64) = 8.00$, $p = 3.7e^{-11}$; L4: $t(64) = 9.66$, $p = 4.1e^{-14}$; L5/6: $t(64) = 7.58$, $p = 1.8e^{-10}$) during the N2pc interval (averaged 150–190 ms following array onset) (*Figure 5—figure supplement 2*).

Importantly, we tested whether the N2pc varied across sessions with or without color-selective columns sampled. We found no difference between N2pc polarization (150–190 ms after the array) between sessions with ($n = 17$) or without ($n = 13$) sampling of color selective columns (two sample $t$ test: $t(28) = -0.75$, $p = 0.46$). This invariance is expected because extracortical EEG spatially integrates signals from multiple cortical columns.

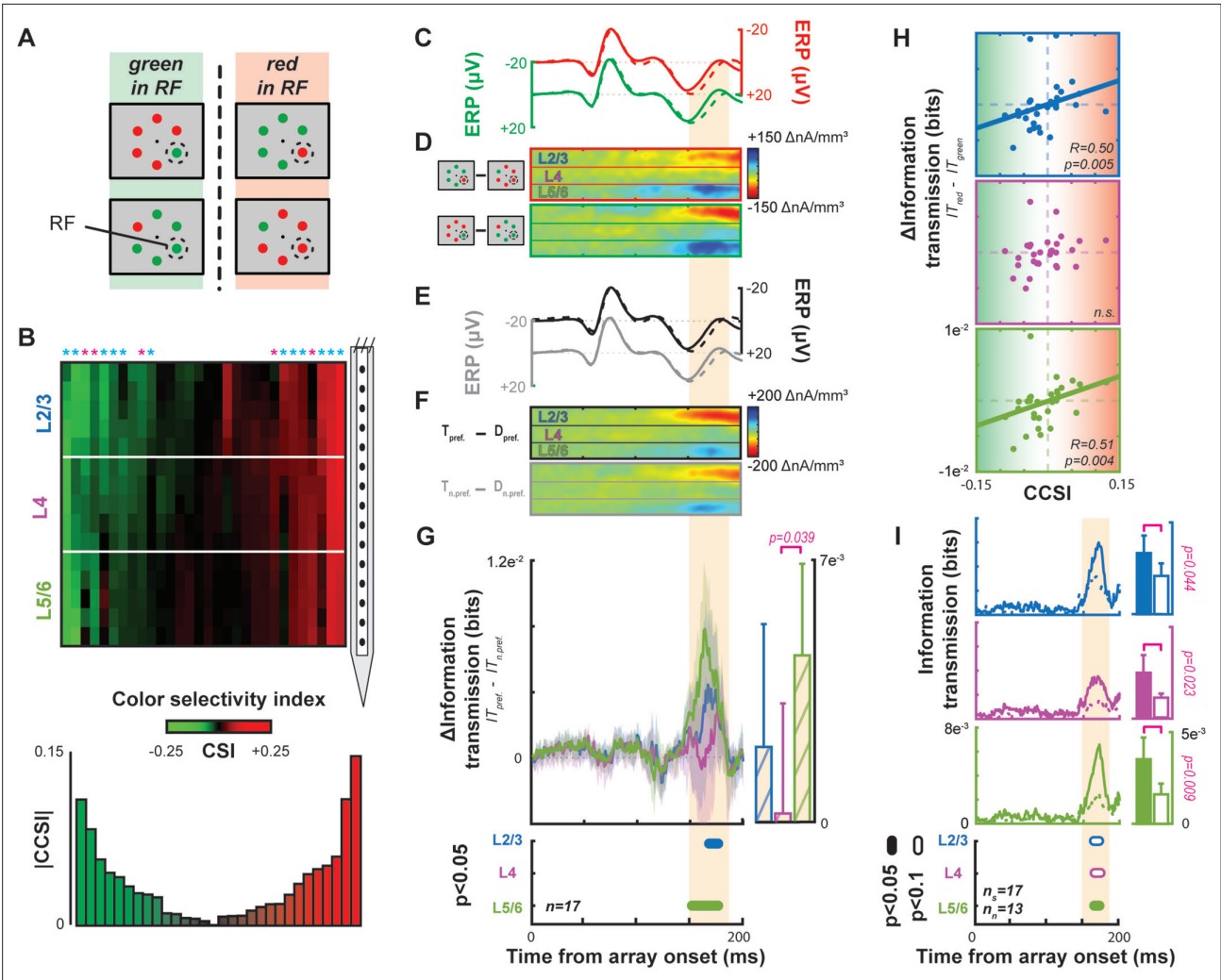

**Figure 5.** Contribution of columnar feature selectivity to the N2pc. Conventions as in *Figure 3*. (**A**) Visual search array configurations used for color selectivity analyses. (**B**) Laminar profiles of red/green color selectivity across all sessions. The hue of each point across cortical depth signifies the value of a color selectivity index (CSI), derived from local gamma power. CSI values < 0 (>0) indicate preference for green (red). CSI is smoothed across adjacent channels for display. Sessions are sorted from left to right based on a column color selectivity index (CCSI) that estimates each column's combined selectivity. A bar plot of session-wise CCSI is plotted below. Asterisks indicate columns with significant color-selectivity (Wilcoxon signed rank, p < 0.05). Asterisk color indicates monkey (Ca cyan; He magenta). (**C**) Average event-related potentials (ERPs) for trials when a red (top) or green (bottom) target or distractor appeared in the receptive field (RF) of the 17 color selective columns. Conventions as in *Figure 3*. (**D**) Difference in current source density (CSD) when the target relative to distractor appeared in the columnar population RF when a red (top) or green (bottom) stimulus appeared in the RF (n = 17). (**E**) Average ERP for trials when the preferred color (top) or non-preferred color (bottom) target or distractor appeared in the RF (n = 17). Conventions as in *Figure 3*. (**F**) Difference in CSD when the target relative to distractor appeared in the RF with the preferred (top) or non-preferred (bottom) color. (**G**) Average difference in information transmission between laminar CSD and N2pc when preferred relative to non-preferred stimulus color appeared in RF. Conventions as before. More information was transmitted when a stimulus of the preferred color appeared in the RF. (**H**) Correlation between difference in information transmission across color columns and CCSI for each session for L2/3 (blue, top), L4 (purple, center), and L5/6 (green, bottom). Spearman correlation reported in lower right of each plot with data from all 30 sessions. Color-specific information transmission scaled with magnitude of color selectivity. (**I**) Information transmission for columns with (solid, n = 17) and without (dashed line, n = 13) feature selectivity for L2/3 (top), L4 (middle), and L5/6 (bottom). Intervals with significant differences are plotted below at two alpha levels for a two-sample t test (filled: 0.05; unfilled: 0.1). Bars plot average with upper 95% confidence interval of information transmission during the N2pc for columns with (left) and without (right) feature selectivity. Significant differences are indicated with a bracket and p value from a two-sample t test.

The online version of this article includes the following figure supplement(s) for figure 5:

**Figure supplement 1.** Single session example (monkey Ca) of the observed difference in information transmission depending on columnar color preference.

**Figure supplement 2.** Attentional modulation is present in cortical columns not selective for an attentional target feature present in the pop-out search task.

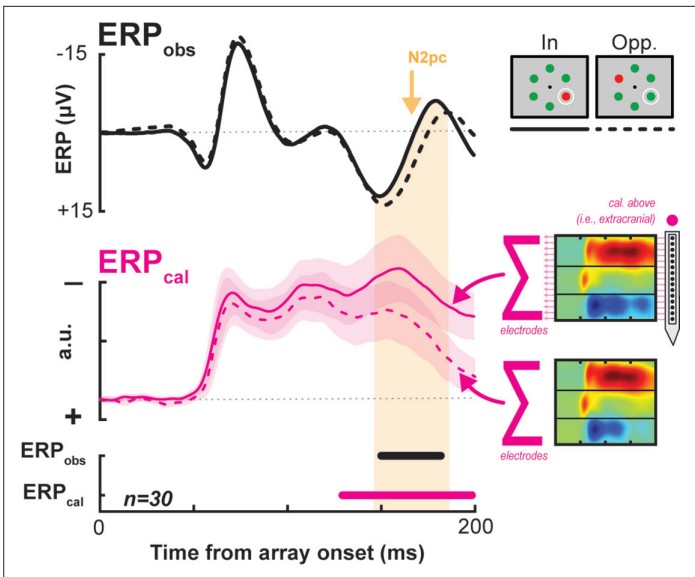

**Figure 6.** Comparing an estimated field potential generated from the current source density (CSD) across the cortical columns to the actually observed extracortical event-related potential (ERP). Black lines indicate the empirically measured event-related potential (ERP_obs, top), averaged across sessions. The pink line indicates the estimated ERP calculated from the synaptic currents across V4 columns, averaged across sessions (ERP_cal, center). Synaptic currents at each electrode are measured and divided by the Euclidean distance of the electrode from the extracortical surface (see Materials and methods; *Nicholson and Llinas, 1971*; *Kajikawa and Schroeder, 2011*). ERP for target present in the receptive field (RF) versus target opposite the RF is shown as solid and dashed lines, respectively (example array for each condition shown at top right). Clouds around ERP_cal lines indicate 95% confidence intervals across sessions for each condition. Note that despite differences in overall waveshape (which are likely due to the fact that V4 is not the only contributor to the attention-independent, visually evoked ERP), the timing of differences within signal types can be compared. The congruence in polarization of the difference in potentials is of similar note.

## Translaminar currents in V4 recapitulate the N2pc

CSD is computed by differentiating between LFPs to eliminate volume-conducted signals that do not arise from local circuit activity. Using an inverse procedure (i.e., summing the CSD), it is possible to estimate the LFP without contamination by volume-conducted activity (*Nicholson and Llinas, 1971*; *Kajikawa and Schroeder, 2011*). We used this approach to compute an estimated extracortical ERP. Specifically, we computed the sum of currents produced by a cortical column to estimate the extra-cortical signal at a position directly above. The resultant potential (ERP_cal) distinguished the target from a distractor in the RF throughout the N2pc (*Figure 6*). In other words, the summed potential generated by currents along V4 columns differentiates between attention conditions simultaneous with the extracortically measured attention-associated signal. Note that the shape of the observed extracortical ERP (ERP_obs) differs from the estimated extracortical ERP_cal. This is expected because the ERP_obs reflects several more variables such as volume-conducted contributions of nearby columns as well as the filtering and attenuating effects of the tissue and cranium above the gray matter (*Nunez and Srinivasan, 2006*). Moreover, the ERP_cal does not reflect the potential contributions of other visual areas. Given these expected differences, it is remarkable how well the difference in ERP_cal predicts the timing of the attention-associated electric field.

## Discussion

Bioelectric potentials have practical and clinical applications when their generators are known. For example, the electrocardiogram is useful in medicine because the physiological process associated with each phase of polarization is understood. Likewise, the electroretinogram is useful because the cell layers associated with each polarization are understood. In contrast, human ERP components

indexing cognitive operations will have limited and only fortuitous utility until their neural generators are known.

The ERP indices of attention such as the N2pc or Pd are commonly used to assess the deployment of attention in human participants, but can also be observed in macaque monkeys, enabling systematic concurrent EEG and intracranial neurophysiological recordings. Our objective was to identify the neural generator of the attention-associated electric fields that comprise ERPs like the N2pc. Using inverse modeling of cranial surface EEG and laminar resolved CSD in a cortical area, we demonstrate that translaminar synaptic currents in visual cortical area V4 contribute to the generation of attention-associated electric fields during visual search. The dipole resulting in this electric field stemmed from layer-specific interactions in extragranular (top-down recipient) cortical layers. Unexpectedly, we discovered that the contribution of a cortical column to the overlying electric field depended on whether the visual feature in the RF matched the selectivity of the column – an important consideration in the mechanism producing EEG potentials that may not be observable through the macroscopic EEG signal alone.

The attention-associated electric field measured in our task is most likely representative of the commonly measured N2pc component of the EEG ERP. Given our findings regarding the functional architecture comprising attention-associated electric fields, it is conceivable that the N2pc arises from multiple, anatomically distinct cortical areas. That is, given the ubiquity of columnar architecture in sensory cortex and the specificity of visual feature representations to different cortical areas, electric dipoles formed across visual cortical layers could come about across multiple visual cortical areas with the relative contribution of each depending on the feature being attended to. This realization could help reconcile conflicting interpretations of the cognitive states and operations that are supposed to be indexed by the N2pc (*Eimer, 1996*; *Kiss et al., 2008*; *Pagano and Mazza, 2012*; *Foster et al., 2020*). It may also help account for variability in the N2pc as a function of attentional target presence in the lower versus upper visual hemifield (*Luck et al., 1997b*) given the positioning of retinotopic representations along the cortical surface (*Gattass et al., 1988*) – a potential focus for future study. Moreover, contributions from areas other than V4 are plausible because previous neurophysiological studies in macaques demonstrate attentional selection signals during visual search in the temporal (e.g., *Sato, 1988*), parietal (e.g., *Bisley and Mirpour, 2019*), and frontal (e.g., *Thompson et al., 2005*; *Zhou and Desimone, 2011*) lobes. Of particular note, neuroimaging studies in humans indicate a contribution to the N2pc from posterior parietal cortex (*Hopf et al., 2000*). In the same vein, FEF neurons locate the target among distractors as early as, or even before, the N2pc arises (*Cohen et al., 2009*; *Purcell et al., 2013*). Given the interconnectivity of FEF and V4 (*Schall et al., 1995*;

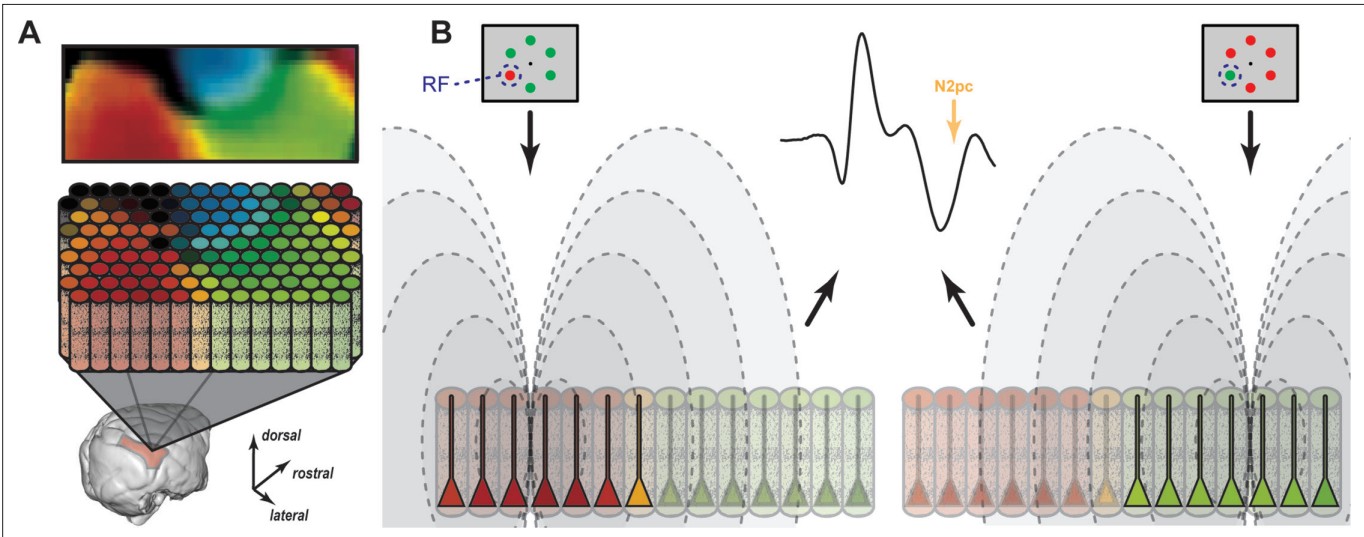

**Figure 7.** Feature mosaic hypothesis. (**A**) A map of preferred color in area V4 derived from optical imaging (*Tanigawa et al., 2010*) with corresponding color columns in area V4. (**B**) Relative contributions of color selective cortical columns to the N2pc when a red (left) or green (right) target appears in the receptive field (RF). Intensity of pyramidal neuron activity is indicated by saturation in the diagram. The mesoscopic columns produce electric fields (dashed lines) that sum to produce the equivalent event-related potential (ERP).

*Ungerleider et al., 2007*; *Gregoriou et al., 2012*; *Ninomiya et al., 2012*), the frontal lobe could thus be the functional origin of an attentional selection signal communicated to V4 and other posterior areas (*Armstrong and Moore, 2007*; *Ekstrom et al., 2009*; *Gregoriou et al., 2009*; *Gregoriou et al., 2012*; *Marshall et al., 2015*; *Popov et al., 2017*), which in turn generate the N2pc (*Westerberg and Schall, 2021a*).

Our discovery that dipoles established by synaptic currents in visual cortical columns underlie the generation of the attention-associated electric fields is consistent with the observation that ERP components such as the N2pc are largest over the occipital lobe in humans (*Luck and Hillyard, 1990*; *Luck and Hillyard, 1994*) and macaques (*Woodman et al., 2007*), which is also observed with MEG (*Hopf et al., 2000*). Our investigation to identify interlaminar interactions producing the N2pc has offered unexpected insights into the underlying neural circuitry. Visual cortical area V4 contains a functional map of hue along its surface (*Tanigawa et al., 2010*) with individual columns comprising the map preferring the same color (*Zeki, 1973*; *Zeki, 1980*; *Tootell et al., 2004*; *Conway and Tsao, 2009*; *Kotake et al., 2009*; *Westerberg et al., 2021b*). We replicated the finding of columns specified by color-feature selectivity and discovered that the contribution of a column to the global electric field was greater for the preferred feature. Specifically, columns that preferred green (or red) contributed more to the electric field when the item in the RF was green (or red) rather than red (or green). This circuitry likely supports the decoding of visual features like color from EEG (*Sandhaeger et al., 2019*; *Sutterer et al., 2021*). The biophysical implications of this unexpected finding are illustrated in *Figure 7*, which portrays how an attention-associated electric field like the N2pc can arise from mosaics of different cortical columns. While all columns with the attended target in their RF contribute to the N2pc, columns with RF enclosing a target with preferred features establishes stronger dipoles than do columns representing other features or visual field locations. If target position or feature change, then other columns contribute the strongest dipoles.

However, these findings leave several unanswered questions. First, the details of the attentional mechanism enacted in the cortical microcircuit and in turn manifested in the EEG remain unknown. In this study, the task design required both spatial and feature-based attention. Existing models of attention put forward hypotheses for the findings reported here. For example, multiplicative gain would predict a larger increase in response for preferred versus non-preferred stimuli when attended in the RF (e.g., *McAdams and Maunsell, 1999*). Alternatively, feature similarity gain would predict that regardless of what is in the RF, red-preferring columns would increase activity and green-preferring would decrease activity when attending red and vice versa (e.g., *Martinez-Trujillo and Treue, 2005*). Further work should be undertaken to disentangle the contributions of spatial and feature attention to this attention-associated field generation. Second, we do not know whether the spatial shifts in the voltage distribution entailed by the mosaic hypothesis can be resolved on the human scalp due to smearing of the signals as they propagate through the skull and scalp (*Nunez and Srinivasan, 2006*). However, given decoding of features from EEG can be achieved (*Sandhaeger et al., 2019*; *Sutterer et al., 2021*), that would suggest some spatial information is preserved. Additionally, we do not know if this observation generalizes to other cortical areas or other ERP components. However, the discovery has this general implication: A given ERP can arise from qualitatively different neural circuit configurations. This implication entails specific limits on the nature of mechanistic inferences available from ERP measures.

Other aspects of the data deserve further consideration. First, our information theoretic analyses, while yielding clear results through rigorous means, produced values that were of noticeably smaller magnitude than what has been reported in previous studies (e.g., *Optican and Richmond, 1987*; *Reich et al., 2001*; *Timme and Lapish, 2018*). We are not concerned about this difference because no previous study has performed these measures on the relationship between intracortical synaptic currents and extracortical electric fields, so we have no strong prior about the magnitude of information theoretic results to expect. Also, the previous information theoretic analyses have been applied most commonly to measure relationships between pairs of single units, but we are comparing mesoscopic currents with macroscopic EEG, which is likely comprised of the activity of many cortical columns beyond that being concurrently measured intracortically.

Second, the polarity of the N2pc measured concurrently with intracortical laminar activity was opposite what has been previously described in macaque monkeys (*Woodman et al., 2007*). We believe this is an unfortunate consequence of differences in the referencing arrangements between

the original and the present study. The previous work sampled EEG from an electrical lead embedded in the outer skull referenced to either linked ears or a frontal, extracranial electrode. We sampled EEG from LMA contact(s) outside the skull referenced to a rod supporting the LMA, which extended into the brain. As such, imbalanced measurement of the electric fields across the putative generator could lead to the inverted ERP polarity. We should also acknowledge possible differences caused by the presence of the craniotomy over the lunate gyrus. This curiosity can be resolved by sampling EEG from the cranial surface before and after a craniotomy with the alternative referencing arrangements. Regardless of the explanation, though, our findings of a strong association between V4 laminar substrates and the N2pc do not depend on the EEG polarization.

As a final note, it is important to consider what comes next for this program of research. Two avenues seem promising. In this study we are limited in that we only observed relationships between otherwise unaltered signals. While causal manipulations to neural circuits in cognitive tasks come with their own limitations (e.g., you are no longer observing the normal functioning system and consequent behavior), much could be gleaned about relative contributions from direct stimulation or inactivation of the putative circuitry generating these electric fields. In a similar vein, a biophysical modeling approach will yield more information on the relationships of attention-associated signals across scales. That is, by now knowing something about the circuit and mechanism yielding these attention-associated electric fields, we are able to use biophysically plausible computational models to gain further insight through simulations. Both approaches seem well suited to build on the findings detailed in this study. Ultimately the goal through these means and beyond should be to bridge the gap between what we know of the neurophysiology of attention at the microscopic scale to human attention-associated signals such as the N2pc.

# Materials and methods

## Key resources table

| Reagent type (species) or resource | Designation | Source or reference | Identifiers | Additional information |
|---|---|---|---|---|
| Biological sample (*Macaca radiata*) | Bonnet macaque; Ca, He | Wake Forest University, NC, USA | | V4 laminar neurophysiology subjects |
| Biological sample (*Macaca radiata*) | Bonnet macaque; P | University of Colorado, CO, USA | | 10/20 EEG subject |
| Biological sample (*Macaca mulatta*) | Rhesus macaque; Z | Lovelace Biomedical: http://www.lovelacebiomedical.org/ | | 10/20 EEG subject |
| Software, algorithm | MATLAB | Mathworks: https://www.mathworks.com/ | | Analysis software |
| Software, algorithm | CURRY | Compumedics Neuroscan: http://www.compumedicsneuroscan.com/ | | Analysis software |
| Software, algorithm | Brainstorm | Brainstorm: http://www.neuroimage.usc.edu/brainstorm | | Analysis software |
| Software, algorithm | TEMPO | Reflective computing: http://www.greatislandsoftware.com/ | | Behavioral control software |
| Other | S-probe | Plexon: http://www.plexon.com/ | | Recording electrode array |
| Other | Electrophysiology equipment; MAP | Plexon: http://www.plexon.com/ | | 10/20 EEG recording system |
| Other | Electrophysiology equipment; RZ2; PZ5 | Tucker-Davis Technologies: http://www.tdt.com/ | | V4 laminar neurophysiology recording system |
| Other | Eye tracker; Eye Link II | SR Research: http://www.sr-research.com/ | | Monocular eye tracking system |
| Other | Ceramic screws | Thomas Recording: http://www.thomasrecording.com/ | | |
| Other | Dental acrylic | Lang Dental: http://www.langdental.com/ | | |
| Other | Recording chamber | Crist Instrument: http://www.cristinstrument.com/ | | |

## Animal care

Procedures were in accordance with National Institutes of Health Guidelines, Association for Assessment and Accreditation of Laboratory Animal Care Guide for the Care and Use of Laboratory Animals, and approved by the Vanderbilt Institutional Animal Care and Use Committee following United States Department of Agriculture and Public Health Services policies. Animals were socially housed. Animals were on a 12 hr light-dark cycle and all experimental procedures were conducted in the daytime. Each monkey received nutrient-rich, primate-specific food pellets twice a day. Fresh produce and other forms of environmental enrichment were given at least five times a week.

## Surgical procedures

Two male macaque monkeys (*Macaca mulatta* monkey Z, 12.5 kg; *Macaca radiata* monkey P, 9 kg) were implanted with head posts and skull-embedded EEG arrays using previously described techniques (*Woodman et al., 2007*). One monkey (monkey P) was implanted with a subconjunctive eye coil. Two male macaque monkeys (*Macaca radiata*; monkey Ca, 7.5 kg; He, 7.3 kg) were implanted with head posts and MR compatible recording chambers with craniotomy over V4. Anesthetic induction was performed with ketamine (10 mg/kg). Monkeys were then catheterized and intubated. Surgeries were conducted aseptically with animals under $O_2$, isoflurane (1–5%) anesthesia. EKG, temperature, and respiration were monitored. Postoperative antibiotics and analgesics were administered. Further detail is documented elsewhere (*Woodman et al., 2007*; *Westerberg et al., 2020a*; *Westerberg et al., 2020b*).

## Magnetic resonance imaging

Anesthetized animals were placed in a 3 T MRI scanner. T1-weighted three-dimensional (3D) MPRAGE scans were acquired with a 32-channel head coil equipped for SENSE imaging. Images were acquired using 0.5 mm isotropic voxel resolution with parameters: repetition 5 s, echo 2.5 ms, flip angle 7 degrees.

## Visual search tasks

Monkeys performed a color pop-out (monkeys Ca, He, and Z) or T/L (monkey P) search. Search arrays were presented on a CRT monitor at 60 Hz, at 57 cm distance. Stimulus generation and timing were done with TEMPO (Reflective Computing). Event times were assessed with a photodiode on the CRT. We used isoluminant red and green disks on a gray background (pop-out) or uniform gray T's and L's on a black background (T/L). Target feature varied within session for monkeys Ca, He, and Z. Monkey P identified the same target on any given session (T or L) but changed specific targets session to session. Trials were initiated by fixating within 1 (monkeys Ca and He) or 2 (monkeys P and Z) degrees of visual angle (dva) of a fixation dot. Time between fixation and array onset was at least 500 ms (monkey P: 500–1000 ms; Z: 500 ms; Ca and He: 750–1250 ms). For monkeys experiencing a range of fixation periods (monkeys Ca, He, P), a nonaging foreperiod function was used to determine the fixation period on a trial-by-trial basis. Arrays comprised of six (monkeys Ca and He) or eight (monkeys P and Z) items. Monkeys P and Z experienced invariable array eccentricity (10 dva) and item size (monkey P: 1.3 × 1.3 dva; Z: 1 × 1 dva). Two items were positioned on the vertical meridian, two on the horizontal, and the four remaining items equally spaced between. Monkeys Ca and He viewed items where size scaled with eccentricity at 0.3 dva per 1 dva eccentricity so that they were smaller than the average V4 RF (*Freeman and Simoncelli, 2011*). The angular position of items relative to fixation varied session to session so that one item was positioned at the center of the RF. Items were equally spaced relative to each other and located at the same eccentricity. Each trial, one array item was different from the others. Monkeys saccaded to the oddball within 1 (monkeys Ca and He) or 2 s (monkeys P and Z) and maintained fixation within 2–5 dva of the target for more than 400 ms (monkeys Ca, He, and Z: 500 ms; monkey P: 400–800 ms). Note that monkeys Ca, P, and Z were trained to versions of their respective search tasks that included catch trials where no target was present and they were tasked to remain fixating. Monkey He did not experience catch trials. Juice reward was administered following successful completion of the trial. The target item had an equal probability of being located at any of the six or eight locations. Eye movements were monitored at 1 kHz or 250 Hz using a corneal reflection system (monkeys Ca, He, and Z) or a scleral search coil (monkey P), respectively. Microsaccades

were detected using an automatic algorithm (*Otero-Millan et al., 2014*). If the monkey failed to saccade to the target, they experienced a timeout (1–5 s).

## 10-20 EEG recordings

Two monkeys with intact skulls (i.e., lacking craniotomies) were implanted with an array of electrodes approximating the human 10–20 system locations (monkey P: FpFz, C3, C4, P3, P4, OL, OR, Oz; monkey Z: FpFz, Fpz, F3, F4, FCz, Cz, C3, C4, Pz, P5, P6, POz, O1, O2, Oz) (*Jasper, 1958*). Referencing was done using either the FpFz electrode (monkey P) or through linked ears (Z). The impedance of the individual electrodes was confirmed to be between 2 and 5 kOhm at 30 Hz, resembling electrodes used for human EEG. EEG was recorded using a Multichannel Acquisition Processor (Plexon) at 1 kHz and filtered between 0.7 and 170 Hz. Data was aligned to array onset and baseline corrected by subtracting the average activity during the 50 ms preceding the array onset from all timepoints. Data was clipped 20 ms prior to saccade to eliminate eye movement artifacts.

## Simultaneous laminar V4 and extracortical recordings

The extracortical electric fields and laminar V4 neurophysiology were acquired at 24 kHz using a PZ5 and RZ2 (Tucker-Davis). Electric signals between 0.1 Hz and 12 kHz were observable with this system. V4 data was collected from two monkeys (monkey Ca: left hemisphere; He: right) across 30 sessions (monkey Ca: 21; monkey He: 9) using 32-channel linear electrode arrays with 0.1 mm inte-relectrode spacing (Plexon) introduced through the intact dura mater each session. Recordings were conducted with the electrode in a tube-grounded, reference-grounded configuration which grounds the stainless-steel support tube of the electrode and grounds the reference of the headstage. Arrays spanned layers of V4 with a subset of electrode contacts deliberately left outside of cortex. The extra-cortical electric field was derived from the most superficial electrode outside the brain (above the dura mater) using the same tube-grounded, referenced-grounded configuration and filtered between 1 and 100 Hz. CSD was computed from the raw signal by first extracting the LFP (signal filtered between 1 and 100 Hz, identical to the extracortical signal) and then taking the second spatial derivative along electrodes (*Nicholson and Freeman, 1975*; *Schroeder et al., 1998*; *Mehta et al., 2000*; *Westerberg et al., 2019*) and converting voltage to current (*Logothetis et al., 2007*). We computed the CSD by taking the second spatial derivative of the LFP:

$$\text{CSD}(t, d) = -\sigma \left( \frac{x(t,d-z) + x(t,d+z) - 2x(t,d)}{z^2} \right)$$

where x is the extracellular voltage at time t measured at an electrode contact at depth d and z is the interelectrode distance and σ is conductivity. Both EEG and CSD were baseline corrected at the trial level by subtracting the average activation during the 300 ms preceding array onset from the response at all timepoints. Extracortical electric field potentials and CSD profiles were clipped 10 ms prior to saccade at the trial level to eliminate the influence of eye movements.

Population spiking was measured and analyzed to supplement primary LFP and CSD results. Multi-unit activity was derived through well-documented means (*Legatt et al., 1980*) and has been demon-strated to be effective across multiple brain areas (*Logothetis et al., 2001*; *Roelfsema et al., 2004*; *Self et al., 2013*; *Shapcott et al., 2016*; *Tovar et al., 2020*; *Westerberg et al., 2020a*; *Xing et al., 2009*). The broadband neural signal was lowpass filtered at 3 kHz, highpass filtered at 300 Hz, full-wave rectified, and lastly, lowpass filtered at 150 Hz. This signal reliably reflect neural population dynamics (*Trautmann et al., 2019*). Additionally, multiunit activity in V4 has been shown to reliably reflect attentional modulation (*Mehta et al., 2000*; *Nandy et al., 2017*).

## Laminar alignment

Orthogonal array penetrations were confirmed online through a reverse-correlation RF mapping procedure (*Nandy et al., 2017*; *Westerberg et al., 2019*; *Cox et al., 2019a*; *Cox et al., 2019b*; *Dougherty et al., 2019*; *Figure 2—figure supplement 1*). RFs were found to represent portions of visual space consistent with previous reports of V4 (*Gattass et al., 1988*). Positions of recording sites relative to V4 layers were determined using CSD (*Schroeder et al., 1998*; *Nandy et al., 2017*). Current sinks following visual stimulation first appear in the granular input layers of cortex, then ascend to the supragranular compartment. Previously described observations of laminar V4 CSD include a sink in the

infragranular layers following the ascent to the supragranular layers (*Nandy et al., 2017*), an observation we do not observe in our data. This is likely because we used task-evoked CSD for alignment with stimulus presentation persisting throughout the measurement interval whereas the descending sink observation was found with very brief stimulus presentations. It is likely that the strength of the persistent supragranular sink is masking the previously reported infragranular sink (*Mitzdorf, 1985*). We computed CSD and identified the granular input sink session-wise. Sessions were aligned by this input sink. 'L4' refers to granular input layer, 'L2/3' – supragranular layers, and 'L5/6' – infragranular layers. Each laminar compartment was assigned the same number of recording sites to alleviate biases during analysis.

## Inverse modeling

Inverse modeling of 10–20 EEG recordings was performed in CURRY 8 (Compumedics Neuroscan). 3D head reconstruction was created for each monkey (P and Z) using the boundary element method (*Hämäläinen and Sarvas, 1989*). This method takes into account individual monkey's surface morphologies to create models of cortex surface, inner and outer skull, and skin boundaries. This model was used in conjunction with EEG to compute a voltage distribution over the cortical surface. We calculated the current density with sLORETA, which calculates a minimum norm least squares that divides current by the size of its associated error bar, yielding F scores of activation. sLORETA produces blurred but accurate localizations of point sources (*Pascual-Marqui, 2002*). Other algorithms such as minimum norm and SWARM were modeled as well, with agreement between models sufficient not to change any conclusions.

## Information theory analyses

Information theory (*Shannon, 1948*) analyses were chosen for several reasons. First, information theory analyses yield results in terms of 'bits' which can be used to directly compare effect sizes across measurement methods (e.g., CSD, extracortical signal, and array composition [directed spatial attention]). Next, these analyses are inherently multivariate and able to capture linear and nonlinear relationships. Furthermore, information theory is model independent and does not necessitate a specific hypothetical structure in order to detect relationships between signals. This combination allows us to detect relationships between the extracortical signal and CSD signal that might not be linear and therefore would not be captured by linear models or correlation analyses. We chose to measure pairwise mutual information and information transmission to gauge the relationships between our three 'signals' (e.g., extracortical, CSD, and array composition [directed spatial attention]). Mutual information is the reduction in uncertainty in one variable afforded by another known variable. That is, mutual information is greater when you know the state of one variable covaries with the state of the other variable. If the two variables do not correspond well, mutual information is low. Therefore, the reduction in uncertainty is formalized as 'information' which is relayed in bits. Mathematically, mutual information is captured by the following equation (*Cover and Thomas, 2006*; *Beer and Williams, 2015*):

$$I(X; Y) = H(X) - H(X|Y)$$

where H*(X)* and H*(X|Y)* are the entropy X and X given Y, respectively. Entropy for a signal (S) is computed by:

$$H(S) = \sum_i p(s_i) \log \frac{1}{p(s_i)}$$

where p(s) is the probability distribution for signal s and i is the signal state. Therefore, mutual information can be computed probabilistically by:

$$I(X; Y) = \sum_i \sum_j p(x_i y_j) \log \frac{p(x_i y_j)}{p(x_i) p(y_j)}$$

where p(x), p(y) are the probability distributions for X and Y, and p(x,y) is the joint probability distribution of X and Y across signal states i and j for X and Y, respectively.

While mutual information describes the relationship between the two signals (for our purpose: CSD and the extracortical signal, CSD and directed spatial attention, or the extracortical signal and directed spatial attention), it does not allow for the evaluation of two signals regarding a third (e.g., CSD and the extracortical signal regarding directed spatial attention). For analyses where we want to understand information regarding the allocation of directed attention from the synaptic currents in V4 to the extracortical signal, we use a modified equation rooted in the same entropy/mutual information principles. In computing information transmission, we are interested in the information about X (directed spatial attention), transferred from Y (CSD in V4) to Z (extracortical signal) formalized as:

$$I_T(X; Y_{past} \rightarrow Z_{future}) = I_{min}(X; Z_{future}, \{Z_{past}, Y_{past}\}) - I_{min}(X; Z_{future}, Z_{past})$$

where past and future describe the timepoints when the data is taken from. The information transmission ($I_T$) is taken as the difference between two minimum information calculations. The minimum information ($I_{min}$) is computed regarding the combination of the individual signals ($S_1$ and $S_2$) at the specified intervals as:

$$I_{min}(X; S_1, S_2) = \sum_x p(x) min \ \{I(X = x; S_1), I(X = x; S_2)\}$$

where p(x) is the probability distribution for signal X and x are the possible states of X. By taking into account different timepoints for the signals, we can interpret this computation as the information about X (directed spatial attention) shared by $Y_{past}$ (e.g., earlier CSD in V4) and $Z_{future}$ (e.g., later extracortical signal) that was not already in $Z_{past}$ (e.g., earlier extracortical signal).

Above information theory analyses were performed using the Neuroscience Information Theory Toolbox (**Timme and Lapish, 2018**). Pairwise mutual information and information transmission were computed at each timepoint across trials for each session using default parameters. Five uniform count bins were used for data binning; 10 ms was used for time lag for information transmission. Only correct trials were included. Information theory measures were computed for each millisecond for the entire interval displayed in each analyses' respective figure panel. CSD for each laminar compartment was computed by taking the average activity of five sites at the trial level included in each laminar compartment. For mutual information between target position and the extracortical signal, target position was binary where target was either contra- or ipsilaterally presented. For computations within V4, target position was binary where target was either in the RF or positioned opposite the RF; 5000 Monte Carlo simulations were used to generate a distribution of null model values which experimental values were compared to ($\alpha = 0.05$).

## Feature selectivity

For each recording site within a column, gamma power (30–150 Hz) (**Maier et al., 2010**) responses were computed when either a red distractor was presented to the RF of the column or when a green distractor was present to the RF. Responses were taken as the average activation 75–200 ms following array onset. An index was computed from these responses by subtracting the 2 and dividing by their sum (CSI). Values were therefore bounded between –1 and 1 where larger magnitude indicates greater selectivity for green (toward –1) or red (toward 1). CCSI was computed as the average of CSIs along the entire column. We performed Wilcoxon signed rank tests on the distribution of CSIs across the recording sites of a given cortical column to determine whether a column was significantly color selective ($\alpha = 0.05$). The selective columns were included in feature selectivity analyses where the preferred color and non-preferred color were defined as the color that elicited greater and lesser responses, respectively.

## Estimating field potential from CSD

We calculated the ERP at arbitrary positions from the measured laminar CSD ($ERP_{cal}$) using a previously described model (**Nicholson and Llinas, 1971**; **Kajikawa and Schroeder, 2011**).

$$ERP_{cal}(d_i, t) = A \sum_i \frac{CSD(d_j, t)}{\sqrt{h^2 + |d_j - d_i|^2}}$$

where $ERP_{cal}$ at depth i ($d_t$) for each timepoint (t) is taken as the sum of CSD at depths j ($d_j$) for each timepoint divided by the Euclidean distance to account for the diminishing impact of local currents on more distant field potentials. The factor A acts only as a scaling factor and we cannot accurately estimate the magnitude of the 1D CSD-derived waveform, so we eliminate this parameter from the calculation. This omission is consistent with previous reports (*Kajikawa and Schroeder, 2011*) and limits our comparisons of observed ERP and $ERP_{cal}$ to only shape. However, magnitude differences can be observed between conditions for $ERP_{obs}$ and $ERP_{cal}$, independently. Also, for our purposes, we set h to 0 as we assume that our observed CSD and the calculated ERP are in the same vertical plane.

## Acknowledgements

This work was supported by NIH through the NEI (P30EY008126, R01EY019882, R01EY008890, R01EY027402) and the Office of the Director (S10OD021771). JAW was supported by fellowships from NEI (F31EY031293 and T32EY007135). The authors would like to thank I Haniff, M Feurtado, M Maddox, S Motorny, D Richardson, L Toy, B Williams, R Williams for technical support. The authors would like to thank B Purcell, P Weigand for collecting data, and S Errington, B Herrera, K Lowe, T Reppert, J Riera, A Sajad, E Sigworth for useful conversations regarding the work.

## Additional information

### Funding

| Funder | Grant reference number | Author |
|---|---|---|
| National Eye Institute | F31EY031293 | Jacob A Westerberg |
| National Eye Institute | P30EY008126 | Alexander Maier<br>Geoffrey F Woodman<br>Jeffrey D Schall |
| National Eye Institute | R01EY019882 | Geoffrey F Woodman<br>Jeffrey D Schall |
| National Eye Institute | R01EY008890 | Jeffrey D Schall |
| National Eye Institute | R01EY027402 | Alexander Maier |
| Office of the Director | S10OD021771 | Alexander Maier<br>Geoffrey F Woodman<br>Jeffrey D Schall |
| National Eye Institute | T32EY007135 | Jacob A Westerberg |

The funders had no role in study design, data collection and interpretation, or the decision to submit the work for publication.

### Author contributions

Jacob A Westerberg, Conceptualization, Data curation, Formal analysis, Funding acquisition, Investigation, Methodology, Visualization, Writing – original draft, Writing – review and editing; Michelle S Schall, Conceptualization, Formal analysis, Investigation, Writing – original draft, Writing – review and editing; Alexander Maier, Conceptualization, Formal analysis, Funding acquisition, Supervision, Writing – original draft, Writing – review and editing; Geoffrey F Woodman, Conceptualization, Funding acquisition, Investigation, Methodology, Supervision, Writing – original draft, Writing – review and editing; Jeffrey D Schall, Conceptualization, Formal analysis, Funding acquisition, Methodology, Supervision, Writing – original draft, Writing – review and editing

### Author ORCIDs

Jacob A Westerberg http://orcid.org/0000-0001-5331-8707
Michelle S Schall http://orcid.org/0000-0003-3631-5541
Alexander Maier http://orcid.org/0000-0002-7250-502X
Geoffrey F Woodman http://orcid.org/0000-0003-0946-9297
Jeffrey D Schall http://orcid.org/0000-0002-5248-943X

## Ethics

Procedures were in accordance with National Institutes of Health Guidelines, Association for Assessment and Accreditation of Laboratory Animal Care Guide for the Care and Use of Laboratory Animals, and approved by the Vanderbilt Institutional Animal Care and Use Committee (Protocol M1700067) following United States Department of Agriculture and Public Health Services policies.

## Decision letter and Author response

Decision letter https://doi.org/10.7554/eLife.72139.sa1
Author response https://doi.org/10.7554/eLife.72139.sa2

---

# Additional files

## Supplementary files

- Transparent reporting form
- Supplementary file 1. Recording setup.
- Reporting standard 1. ARRIVE checklist.

## Data availability

Data supporting the findings documented in this study are freely available online through Dryad at https://doi.org/10.5061/dryad.djh9w0w15.

The following dataset was generated:

| Author(s) | Year | Dataset title | Dataset URL | Database and Identifier |
|---|---|---|---|---|
| Westerberg JA, Schall M, Maier A, Woodman G, Schall J | 2021 | Data from: Laminar microcircuitry of visual cortex producing attention-associated electric fields | http://dx.doi.org/10.5061/dryad.djh9w0w15 | Dryad Digital Repository, 10.5061/dryad.djh9w0w15 |

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
