## [Editor Report]

By combining rare EEG and laminar recordings in monkeys, Westerberg and colleagues studied the neural correlates of the well-known attention-related N2pc signal and found that it is due to the activation of extra-granular layers of cortex. Further, this effect was stronger for columns that were more feature selective. These findings are extremely important and a unique contribution to the literature on the neurobiology of attention.

---

## [Decision Letter]

**Decision letter after peer review:**

Thank you for submitting your article "Laminar microcircuitry of visual cortex producing attention-associated electric fields" for consideration by *eLife*. Your article has been reviewed by 3 peer reviewers, one of whom is a member of our Board of Reviewing Editors, and the evaluation has been overseen by Chris Baker as the Senior Editor. The following individuals involved in review of your submission have agreed to reveal their identity: Steven J. Luck (Reviewer #2); Anirvan S Nandy (Reviewer #3).

Essential revisions:

1. Effects of attention in V4 generally start earlier (~100 ms). It is unclear why no effect is observed during earlier time periods in these data. To make better comparison with previous studies (such as Nandy et al., 2017), the authors should show the average PSTHs in supragranular, granular and infragranular layers during both target-out versus target-in conditions. Interestingly, Nandy and colleagues found largest changes in firing rates in the granular layer. To better understand the ERP outside the cortex, the authors should also show the average LFPs in the three layers, for target-in and target-out conditions. It is surprising that MI analysis reveals no significant information about the target in granular layer – given that some attentional effects are seen in upstream areas such as V1 and V2.

2. Eye position analysis: my understanding is that the animals could make a saccade as soon as the arrays were displayed. Given that the main effect of attention is observed after ~150-200 ms, the potential effect of saccade preparation could be important. There could also be small eye movements before the saccade. Given that the RFs were quite foveal for one monkey and not too far from the fixation window, and the effect of attention appears to be quite late, detailed analysis of eye position and microsaccades is needed to rule out the possibility of differences in eye movements between target in and target-out conditions influencing the results. A timeline and some analysis of eye movement patterns would be appropriate. The authors should also clearly mention the mean and SD of the saccade onset.

3. Attention studies typically keep the stimulus in the RF the same to tease out the effect of attention from stimulus selectivity. Ideally, the comparison should be between the two green (or red) in RF conditions as shown in Figure 4A. However, these results are shown only after pooling across all color selective columns. This comparison should be shown from Figure 2 itself (i.e., Figure 2C should have green in the RF and red target outside).

4. Information has been well characterized in a large number of previous studies (generally yielding values between a few bits/s, see for example, Reich et. al, 2001, JNP). Here, the absolute value of mutual information seems rather low. This may be due to the way the information is computed. A discussion about these reasons would be useful for scientists interested in information-theoretic measures.

5. Dependence on feature preference: The effect of spatial and feature attention is well studied. (A multiplicative gain model of spatial attention would predict a larger increase in firing rates and perhaps other signals such as CSD) for preferred versus non-preferred signals. Feature similarity gain model would predict the red preferring columns to increase their activity and green preferring columns to reduce their activity when the animal is attending to the feature red, irrespective of which stimulus is in the receptive field. Here, the task is a pop-out task which likely has both a spatial and feature attention component. The authors should discuss their findings in these contexts. Further, the authors should discuss whether their findings could just be a reflection of the magnitude of the change (which could be larger for preferred versus non-preferred stimulus). The information-theoretic measure should ideally not depend on the absolute magnitude, but these quantities often get biased in non-trivial ways based on the magnitude. Does information transmission depend on the magnitudes of firing rates/CSDs?

6. For columns that were not feature selective, is there an effect of attention? Does the magnitude of N2pc change depend on color selectivity? I think that should be the case based on Figure 4H and 4I, but a plot and/or some quantification would be useful.

7. The most challenging aspect of the study is to provide a solid link from the intracortical activity to the voltage on the cortical surface, and then to the monkey scalp ERPs, and finally to human ERPs. Toward that end, the present study relied entirely on correlational evidence, rather than experimental manipulations. That's quite appropriate for a first step, but it must be considered an important limitation on the conclusions that can be drawn. It would be wonderful if future research took the next step of providing experimental evidence.

8. There are also some troubling aspects of the existing evidence. The scalp ERP effect in this study, and the prior work from this group, is a positive voltage over the contralateral hemisphere, whereas in humans the voltage is negative. This may well reflect the orientation of the relevant cortical surface in monkeys versus humans. However, the voltage on the cortical surface in the present study was negative contralateral to the target, not positive. Unless this opposite voltage on the cortical surface relative to the scalp reflects something about the reference site for the cortical surface electrode, then this makes it difficult to link the intracortical effects and cortical surface effects to the scalp ERP effects. Also, the CSD was negative in the upper layers and positive in the lower layers, again suggesting that the voltage should be negative contralateral to the target on the surface. Ironically, this polarity is what would be expected from the human brain, where a contralateral negativity is observed. The oddity seems to be the contralateral positivity in the monkey scalp data. Also, the cortical surface voltage exhibits a polarity reversal at approximately 180 ms, which is not seen in the intracortical CSD. One possible explanation for the discrepancy is that the scalp voltage likely comes from multiple brain areas besides V4. If, for example, areas on the ventral surface of the occipital and temporal lobes produce stronger scalp voltages than V4 under the present conditions, the opposite orientation of these areas relative to the cortical surface would be expected to produce a positive voltage at the scalp electrodes. The manuscript notes that multiple areas probably contribute to the scalp ERPs and argues that the pattern of intracortical CSD results obtained in V4 will likely generalize to those areas. That seems quite plausible. Moreover, the results are interesting independent of their link to scalp ERPs. Thus, the present results are important even if the scalp polarity issue cannot be definitively resolved at this time.

9. There are also some significant concerns about the filters. The high-pass cutoff was high enough that it could have produced artifactual opposite polarity deflections in the data. If causal filters were applied (e.g., in hardware during the recordings), these artifactual deflections would have been after rather than before the initial deflection, possibly explaining the polarity reversal at 180 ms. If noncausal filters were applied in software, this would be a larger problem and could produce artifacts at both the beginning and end of the waveform. Moreover, the filters were different for the CSD data and the extracortical voltages, which is somewhat problematic for the information theoretic comparisons of these two data sources (but is likely to reduce rather than inflate the effects).

The filter for the intracranial recordings was listed as "0.1-12kHz". Was the high pass cutoff really at 0.1 kHz (100 Hz), or was it supposed to be 0.1 Hz to 12 kHz? A cutoff at 100 Hz would make it impossible to see field potentials corresponding to the N2pc. For the extracortical electrode, the 1 Hz cutoff is still quite high. I think you'd need to show how it impacts an N2pc-like artificial waveform (e.g., one half cycle of a 5 Hz sine wave) so that the effects of the filter on the observed data can be estimated. Also, the authors might want to apply offline filters so that the same effective bandpass is used for the extracortical voltage and the intracortical CSD. (This could be shown in a supplemental figure.)

10. The method section states correctly that "current sinks following visual stimulation first appear in the granular input layer of the cortex, then ascend and descend to extra granular compartments". However in the example CSDs shown in Figure 2, Figure 3, Figure S3 there is no visible current sink in the infra-granular layers. Instead, the identified infra-granular layers show a prolonged current source (e.g. Figure S5B,C), which is unexpected. Can the authors comment on this discrepancy?

11. The example RF profile shown in Figure S5A, although aligned, looks a little strange in that the RFs taper off rapidly in the infra-granular layer. Is this the best representative example? It will be important to see other examples of RF alignment.

12. The study used LFP power in the gamma range to compute the response ratio between red and green stimuli. LFPs measured across the cortical depth are highly correlated, and so would gamma power estimated from the LFPs. Given this, how meaningful is the laminar analysis shown in Figure 4B? How confidently can it be established that the LFP derived gamma power estimates have laminar specificity?

---

## [Author Response]

Essential revisions:1. Effects of attention in V4 generally start earlier (~100 ms). It is unclear why no effect is observed during earlier time periods in these data. To make better comparison with previous studies (such as Nandy et al., 2017), the authors should show the average PSTHs in supragranular, granular and infragranular layers during both target-out versus target-in conditions. Interestingly, Nandy and colleagues found largest changes in firing rates in the granular layer. To better understand the ERP outside the cortex, the authors should also show the average LFPs in the three layers, for target-in and target-out conditions. It is surprising that MI analysis reveals no significant information about the target in granular layer – given that some attentional effects are seen in upstream areas such as V1 and V2.

We have created a new figure showing multiunit activity and LFP across the layers in both attention conditions. It is included here for convenience. Accompanying text has been added to the Results and Discussion sections to address the reviewers’ comments.

The timing of differentiation between attended and unattended in the population spiking activity is evident in both MUA and LFP. We note that the largest magnitude difference in population spiking between attention conditions was observed in the middle layers, consistent with Nandy et al., 2017. We wish to highlight two observations.

First, with respect to the timing of attentional modulation, it should be noted that the attention task used in our study (pop-out visual search) is different from that used by Nandy et al., 2017, Neuron (cued change detection). The timing of “effects of attention” vary according to stimulus properties and task demands (the number of publications demonstrating this is too long to list). Hence, we do not expect equivalence between the times we measure and times Nandy et al. measure. Nonetheless we are happy to include the requested supplementary figure with that caveat in mind.

Second, with respect to the surprising observation of a relationship between activity in the granular layer and the extracortical signal, we think it is important to remember that these information theoretic analyses are not simply correlational. That is, attentional modulation might be observed in both signals, but if the covariation of these signals trial-to-trial does not exist, then we would not expect a relationship in the mutual information analysis.

2. Eye position analysis: my understanding is that the animals could make a saccade as soon as the arrays were displayed. Given that the main effect of attention is observed after ~150-200 ms, the potential effect of saccade preparation could be important. There could also be small eye movements before the saccade. Given that the RFs were quite foveal for one monkey and not too far from the fixation window, and the effect of attention appears to be quite late, detailed analysis of eye position and microsaccades is needed to rule out the possibility of differences in eye movements between target in and target-out conditions influencing the results. A timeline and some analysis of eye movement patterns would be appropriate. The authors should also clearly mention the mean and SD of the saccade onset.

The reviewer makes a valuable observation. Saccades will influence the electrical signals, something we are quite familiar with (e.g., Godlove et al., 2011*,* J Neurophysiol). In an effort to combat this, we have two points worth noting. First, as was the case in the initial submission (which remains the same in the revision), we have clipped signals on a trial-by-trial basis prior to eye movements. By doing so, we cannot have an influence of the motor-related polarization of the task-demanded eye movement on the data.

Second, we have prepared a microsaccade analysis – and accompanying newly added supplementary figure included here for convenience – to determine whether they might be driving the results. To do this, we identified trials where microsaccades occurred using a well-regarded microsaccade detection algorithm (Otero-Millan et al., 2014, J Vis). We then reperformed the information theoretic analysis across sessions after removing trials where microsaccades were detected. Briefly, we found that the information theoretic relationship persists in the absence of trials where microsaccades occurred. We believe this serves as evidence that microsaccades are not responsible for the information theoretic findings.

To address the reviewer’s last point, we have included response time data (defined as the saccade onset latency) in the Results.

3. Attention studies typically keep the stimulus in the RF the same to tease out the effect of attention from stimulus selectivity. Ideally, the comparison should be between the two green (or red) in RF conditions as shown in Figure 4A. However, these results are shown only after pooling across all color selective columns. This comparison should be shown from Figure 2 itself (i.e., Figure 2C should have green in the RF and red target outside).

We have clarified prior to Figure 4 that we used all trials including both colors in each of the attention conditions. That is, while the cartoon in Figure 2 shows only green-attended and red-unattended conditions, green-unattended and red-attended conditions were also included in this analysis. As the proportion of red-target and green-target trials was matched, this first analysis was designed in such a way that the influence of stimulus color should be minimized, yet all trials could still contribute to the calculation. We have included a new supplementary figure (included here for convenience) which is what we believe the reviewer requests. In this addition, we perform the information theoretic computation on only stimulus matched conditions. Briefly, we find that this approach does not seem to alter the temporal profile of information theoretic findings.

4. Information has been well characterized in a large number of previous studies (generally yielding values between a few bits/s, see for example, Reich et. al, 2001, JNP). Here, the absolute value of mutual information seems rather low. This may be due to the way the information is computed. A discussion about these reasons would be useful for scientists interested in information-theoretic measures.

We agree that the exact magnitude of our information theoretic analyses in curious. And while these methods have been widely characterized – they have not been characterized, to our knowledge, in relating intracortical laminar currents to extracortical field potentials. As such, we do not have a strong prior as to what we should expect magnitude-wise. We have expanded the discussion to note this observation and provide potential reasons as to why this might be the case. The conclusion being that further application of these methods to these datatypes is necessary to really gain a fuller sense of what should and shouldn’t be expected.

5. Dependence on feature preference: The effect of spatial and feature attention is well studied. (A multiplicative gain model of spatial attention would predict a larger increase in firing rates and perhaps other signals such as CSD) for preferred versus non-preferred signals. Feature similarity gain model would predict the red preferring columns to increase their activity and green preferring columns to reduce their activity when the animal is attending to the feature red, irrespective of which stimulus is in the receptive field. Here, the task is a pop-out task which likely has both a spatial and feature attention component. The authors should discuss their findings in these contexts. Further, the authors should discuss whether their findings could just be a reflection of the magnitude of the change (which could be larger for preferred versus non-preferred stimulus). The information-theoretic measure should ideally not depend on the absolute magnitude, but these quantities often get biased in non-trivial ways based on the magnitude. Does information transmission depend on the magnitudes of firing rates/CSDs?

The relationship of these findings to the specificities of attentional mechanisms and models is indeed intriguing. As the reviewer suggested, this task likely engages both spatial and feature attention – however, the design was not such that they can be disentangled wholly. We have added text to the Discussion to reflect this consideration. As for the potential influence of response magnitude changes on the information theoretic analyses – the exact parameters were chosen to mitigate concerns about magnitude. That is, we chose a uniform count binning procedure on the data which eliminates potential issues such as outliers driving relationships as well as the changes in variability associated with increases in magnitude. Moreover, the uniform count binning procedure results with states rather than magnitudes which again mitigates response-magnitude-driven effects.

6. For columns that were not feature selective, is there an effect of attention? Does the magnitude of N2pc change depend on color selectivity? I think that should be the case based on Figure 4H and 4I, but a plot and/or some quantification would be useful.

These questions have been addressed in a newly added supplementary figure as well as quantification in the Results. Briefly, we did find an effect of attention non-selective columns. Also, we found the magnitude of N2pc did not depend on color-selectivity of the intracortical recording. The results were reported as:

“We also tested whether feature selective columns, on average, transmitted more information than their non-feature-selective counterparts. […] This invariance is expected because extracortical EEG spatially integrates signals from multiple cortical columns.”

7. The most challenging aspect of the study is to provide a solid link from the intracortical activity to the voltage on the cortical surface, and then to the monkey scalp ERPs, and finally to human ERPs. Toward that end, the present study relied entirely on correlational evidence, rather than experimental manipulations. That's quite appropriate for a first step, but it must be considered an important limitation on the conclusions that can be drawn. It would be wonderful if future research took the next step of providing experimental evidence.

We appreciate the reviewer noting that this manuscript is a valuable step in linking attention-associated electrophysiological signals across species. We also recognize that there is much work to be done in this domain. As requested, we have added to the Discussion the limitation of this type of study as well as what should be considered valuable next steps in this program of research.

8. There are also some troubling aspects of the existing evidence. The scalp ERP effect in this study, and the prior work from this group, is a positive voltage over the contralateral hemisphere, whereas in humans the voltage is negative. This may well reflect the orientation of the relevant cortical surface in monkeys versus humans. However, the voltage on the cortical surface in the present study was negative contralateral to the target, not positive. Unless this opposite voltage on the cortical surface relative to the scalp reflects something about the reference site for the cortical surface electrode, then this makes it difficult to link the intracortical effects and cortical surface effects to the scalp ERP effects. Also, the CSD was negative in the upper layers and positive in the lower layers, again suggesting that the voltage should be negative contralateral to the target on the surface. Ironically, this polarity is what would be expected from the human brain, where a contralateral negativity is observed. The oddity seems to be the contralateral positivity in the monkey scalp data. Also, the cortical surface voltage exhibits a polarity reversal at approximately 180 ms, which is not seen in the intracortical CSD. One possible explanation for the discrepancy is that the scalp voltage likely comes from multiple brain areas besides V4. If, for example, areas on the ventral surface of the occipital and temporal lobes produce stronger scalp voltages than V4 under the present conditions, the opposite orientation of these areas relative to the cortical surface would be expected to produce a positive voltage at the scalp electrodes. The manuscript notes that multiple areas probably contribute to the scalp ERPs and argues that the pattern of intracortical CSD results obtained in V4 will likely generalize to those areas. That seems quite plausible. Moreover, the results are interesting independent of their link to scalp ERPs. Thus, the present results are important even if the scalp polarity issue cannot be definitively resolved at this time.

We thank the reviewer for expressing that the results are important whether this polarity difference can be resolved. This is an interesting observation and quite important to consider carefully. First, it is worth reiterating that the referencing setup in our ‘10/20’ monkeys was different than that for the monkeys where intracranial recordings took place. Specifically, the 10/20 recordings were more similar to our previous reports of monkey EEG (e.g., Woodman et al., 2007, PNAS; Cohen et al., 2009, J Neurophysiol; Purcell et al., 2013, J Neurophysiol). Recordings from these monkeys used either a frontal EEG electrode (approximately FpFz) or linked ears for referencing. These yielded the positive-going N2pc and contrast the negative-going N2pc found in humans. The V4 laminar recordings – and their accompanying extracortical signal – used a different referencing setup that we believe is the most likely candidate for the observed difference. Specifically, these recordings used a tied ground-reference setup which incorporated the support rod of the linear multielectrode array. This support rod extended into the brain meaning we had a neural tissue grounded signal and that the reference spanned the neural generator. Therefore, if we are not measuring both sides of the electric field across the generator equally, we can observe an inverted signal. Unfortunately, we cannot observe the 10/20 EEG distribution with an intracranial reference. Ideally, this could be resolved by an experiment where referencing setups are tested before and after performing craniotomy with a series of reference locations used to understand where exactly this flipping of polarization takes place. We have added this consideration to the Discussion and more thoroughly detailed the referencing setups in the Methods.

9. There are also some significant concerns about the filters. The high-pass cutoff was high enough that it could have produced artifactual opposite polarity deflections in the data. If causal filters were applied (e.g., in hardware during the recordings), these artifactual deflections would have been after rather than before the initial deflection, possibly explaining the polarity reversal at 180 ms. If noncausal filters were applied in software, this would be a larger problem and could produce artifacts at both the beginning and end of the waveform. Moreover, the filters were different for the CSD data and the extracortical voltages, which is somewhat problematic for the information theoretic comparisons of these two data sources (but is likely to reduce rather than inflate the effects).The filter for the intracranial recordings was listed as "0.1-12kHz". Was the high pass cutoff really at 0.1 kHz (100 Hz), or was it supposed to be 0.1 Hz to 12 kHz? A cutoff at 100 Hz would make it impossible to see field potentials corresponding to the N2pc. For the extracortical electrode, the 1 Hz cutoff is still quite high. I think you'd need to show how it impacts an N2pc-like artificial waveform (e.g., one half cycle of a 5 Hz sine wave) so that the effects of the filter on the observed data can be estimated. Also, the authors might want to apply offline filters so that the same effective bandpass is used for the extracortical voltage and the intracortical CSD. (This could be shown in a supplemental figure.)

In revisiting the description of the recording system and filters, we see how some information was conveyed poorly. The language describing the recording in the original submission suggested that online filters were applied to the data as it was being recorded. This was not the case. We have changed that language so that it reads as the data was being collected at a sampling frequency sufficient to observe data between 0.1 Hz and 12 kHz rather than the data being filtered between 0.1 Hz and 12 kHz. Related, the reviewer is correct on the typo regarding units. We did mean 0.1 Hz rather than 0.1 kHz; this has been corrected in the revision. Also, it appears that the description of the processing sequence regarding CSD was ambiguous in the original submission. The CSD underwent the same offline, bandpass filtering procedure (1-100 Hz) as the extracortical signal. We have clarified the Methods accordingly.

Additionally, we have conducted the filtering simulation to see how our filter choice affects a signal representative of the N2pc – as suggested by the reviewer. We have included the result as Author response image 1. Using the same filtering parameters used throughout the manuscript on a 5 Hz half sine wave, we find that the filtering process modestly attenuates the frequency band and period where we expect the N2pc to exist. We have plotted the figure with similar conventions to the manuscript so that the time period and polarization component representative of the N2pc is highlighted.

**Author response image 1. sa2fig1:** 

10. The method section states correctly that "current sinks following visual stimulation first appear in the granular input layer of the cortex, then ascend and descend to extra granular compartments". However in the example CSDs shown in Figure 2, Figure 3, Figure S3 there is no visible current sink in the infra-granular layers. Instead, the identified infra-granular layers show a prolonged current source (e.g. Figure S5B,C), which is unexpected. Can the authors comment on this discrepancy?

We have clarified the Methods to reflect the observations of our data and why they may differ from previous reports. We believe the discrepancy is likely due to the stimulus conditions used to evoke the CSD profile. Specifically, the descending infragranular sink in visual cortical columns has most commonly been described when CSD was computed while monkeys view briefly presented flashes or stimuli (e.g., Schroeder et al., 1998, Cereb Cortex). However, our study uses task evoked CSD to perform the alignment. Importantly, this means there is a persistent stimulus in the receptive field. We believe this persistent stimulus, rather than a flashed stimulus, leads to a persistent, strong sink in the superficial layers of cortex which would mask any current sink present in the infragranular layers (Mitzdorf, 1985, Physiol Rev). This is an observation we made in previous reports (Task evoked CSD: Westerberg et al., 2019, J Neurophysiol vs. Flash evoked CSD: Maier et al., 2010, Front Syst Neurosci), albeit in V1 instead of V4. Given the latency offset between putative granular and supragranular sinks, that we observe receptive fields below the putative granular input sink, and the demonstrable multiunit activation as indicated by the newly included Figure S2, we have no reservations in our assessment of the position of the electrode relative to the layers across sessions.

11. The example RF profile shown in Figure S5A, although aligned, looks a little strange in that the RFs taper off rapidly in the infra-granular layer. Is this the best representative example? It will be important to see other examples of RF alignment.

The attenuation observed in the lower layers is largely due to overall decreased gamma power in the lower layers of cortex as compared to upper and middle layers (Maier et al., 2010, Front Syst Neurosci). At the reviewer’s request, we have added an additional panel to the noted supplementary figure which shows additional laminar receptive field profiles using the evoked LFP so that they are more directly comparable to those shown in Nandy et al., 2017, Neuron.

12. The study used LFP power in the gamma range to compute the response ratio between red and green stimuli. LFPs measured across the cortical depth are highly correlated, and so would gamma power estimated from the LFPs. Given this, how meaningful is the laminar analysis shown in Figure 4B? How confidently can it be established that the LFP derived gamma power estimates have laminar specificity?

An astute observation – there are two aspects to consider. The existence of color-feature columns has been well-documented in V4 (e.g., Zeki, 1973, Brain Res; Zeki, 1980, Nature; Tootell et al., 2004, Cereb Cortex; Conway and Tsao, 2009, PNAS; Kotake et al., 2009, J Neurophysiol; Westerberg et al., 2021, PNAS). This manuscript did not need the evaluation of interlaminar differences in color selectivity to address the question at hand – the top of Figure 4B only serves as a step to the bottom of Figure 4B which provides the measurements used for the subsequent analyses. Thus, the estimation of color selectivity from gamma was sufficient to capture a general sense of the color selectivity of the column. Second, we recently published a manuscript which directly addresses the laminar specificity of gamma with respect to feature selectivity. Westerberg et al., 2021, PNAS uses a spatially localized form of gamma to evaluate color-feature selectivity along V4 columns. In that manuscript, we find a high degree of consistency along the layers of cortex using the gamma signal. Notably, we compared the gamma signal to the population spiking and found a high degree of coherence between selectivity in those two measures as a function of cortical depth. Given the secondary nature of the interlaminar feature selectivity to this submitted manuscript and the detailed report of laminar feature selectivity using the same dataset in another manuscript, we are inclined to leave the analysis reported here as is with adjustments to the text that note these considerations now included in the Results.